# GENBEN: A GENARATIVE BENCHMARK FOR LLM-AIDED DESIGN

## ABSTRACT

This paper introduces GenBen, a generative benchmark designed to evaluate the capabilities of large language models (LLMs) in hardware design. With the rapid advancement of LLM-aided design (LAD), it has become crucial to assess the effectiveness of these models in automating hardware design processes. Existing benchmarks primarily focus on hardware code generation and often neglect critical aspects such as Quality-of-Result (QoR) metrics, design diversity, modality, and test set contamination. GenBen is the first open-source, generative benchmark tailored for LAD that encompasses a range of tasks, from high-level architecture to low-level circuit optimization, and includes diverse, silicon-proven hardware designs. We have also designed a difficulty tiering mechanism to provide fine-grained insights into enhancements of LLM-aided designs. Through extensive evaluations of several state-of-the-art LLMs using GenBen, we reveal their strengths and weaknesses in hardware design automation. Our findings are based on 10,920 experiments and 2,160 hours of evaluation, underscoring the potential of this work to significantly advance the LAD research community. In addition, both GenBen employs an end-to-end testing infrastructure to ensure consistent and reproducible results across different LLMs. The benchmark is available at https://anonymous.4open.science/r/GENBEN-2812.

## 1 INTRODUCTION

Modern circuit design is a complex, multidisciplinary endeavor that demands expertise in numerous areas, including architecture design, performance modeling design space exploration, register-transfer level (RTL) implementations, design verification, physical layout, etc. (Rabaey et al., 2002; Hennessy & Patterson, 2017; Bergeron, 2012). As hardware complexity increases, so too does the overhead associated with design and verification processes, subsequently lengthening the design iteration cycles (Calhoun et al., 2008). Traditional methodologies, which rely heavily on manual implementations in Verilog, are being improved by Chisel (Thomas et al., 1989; Bachrach et al., 2012) and High-Level Synthesis (HLS) (Coussy & Morawiec, 2010; Gajski et al., 2012) that aim to automate RTL code generation by introducing additional abstraction layers. However, even with these advancements, the verification overhead remains labor-intensive. Consequently, there is a growing need for advanced agile hardware design approaches to accelerate hardware development iterations.

With the rise of transformer-based large language models (LLMs) (Zhao et al., 2023; Winata et al., 2021; Chakrabarty et al., 2023), has opened new avenues for hardware design automation. Models like GPT-4(OpenAI, 2023), Claude (Team, 2023), and LLaMA (Touvron et al., 2023a;c; Dubey et al., 2024) have demonstrated promising results not only in natural language processing but also in programming. Within this new paradigm of LLM-Aided Design (LAD) (ICCAD-Committee, 2023; ACM-SIGDA, 2024; Huang et al., 2024), models such as WizardCoder (Luo et al., 2023) and Code-LLaMA (Roziere et al., 2023) have demonstrated significant capabilities.

Building on these advanced models, techniques like fine-tuning (Wei et al., 2021) and retrieval-augmented generation (RAG) (Lewis et al., 2020; Gao et al., 2023) have led to the development of domain-specific models and operational architectures such as GPT4AIGChip (Fu et al., 2023), AutoChip (Thakur et al., 2023c), ChatChisel (Liu et al., 2024b), and ChatCPU (Wang et al., 2024).

These efforts have demonstrated automated hardware design capability using LLMs. This paradigm shift heralds a new wave of innovation in hardware design automation.

To accurately assess the efficacy of hardware code generations, several benchmarks have been introduced, such as RTLLM (Lu et al., 2024), Verigen (Thakur et al., 2023a), and VerilogEval (Liu et al., 2023). As these benchmarks are open-source on GitHub and typically consist of static tests, they can inadvertently be incorporated into training datasets, leading to misleading test results. Moreover, there is a pressing need for improvements in verification coverage, evaluation metrics, and data diversity. For instance, the tests in these benchmarks are relatively simple and unimodal, focusing primarily on syntax and functional pass rates. This focus neglects critical metrics such as synthesizability, debugging capabilities, and performance, power, and area (PPA)(Marakkalage et al., 2024) statistics, which are essential for a comprehensive evaluation.

To address these limitations, we introduce GenBen, an innovative benchmark for systematic evaluation of generative AI capabilities in hardware design. GenBen distinguishes itself from existing works with the following key innovative enhancements:

- **Enhanced Verification Coverage:** We rigorously employ a standard, end-to-end verification flow to maximize the functional coverage of the developed testbench, that maps the generated stimuli to each function point of the RTL design.

- **Diverse and Difficulty Tiering Dataset:** GenBen showcases a multi-source, multimodal, and difficulty-tiered evaluation framework consisting of 300 tests derived from silicon-proven designs, textbooks, StackOverflow, and other sources. Each test is categorized into one of three distinct difficulty levels (L1 to L3), allowing for the fine-grained and targeted enhancement of LLM capabilities in hardware designs.

- **Generative Benchmark Against Data Contamination:** GenBen is a generative benchmark that incorporates both static and dynamic perturbations to distinguish each test from its source dataset. Additionally, we utilize a script-based generation approach to impede automated RTL code extraction by GitHub crawlers, effectively minimizing the risk of test set data leakage.

- **Enhanced Evaluation Metrics:** GenBen incorporates diverse metrics to comprehensively evaluate the generated designs, including the basic syntactical/functional correctness, and Quality-of-Results(QoR)(Yu et al., 2018) metrics like synthesizability, power consumption, area utilization, timing performance, etc.

- **End-to-End Open-Source Workflow:** GenBen integrates tools like Icarus Verilog(Williams, 2023), OpenLane EDA flow(Ghazy & Shalan, 2020), and Open-PDK(Edwards, 2023) to simplify the reproducibility.

The remainder of this paper is organized as follows: Section 2 presents the motivation behind GenBen and reviews related work. Section 3 introduces GenBen architecture and workflow. Section 4 evaluates diverse LLMs using GenBen, and Section 5 concludes this paper.

## 2 RELATED WORKS

To further elucidate the necessity and impact of GenBen in advancing hardware design automation, it is imperative to examine the current state of LLM-aided design (LAD) and the benchmarks used to evaluate such systems. The following sections delve into the integration of LLMs in hardware design and critically analyze the benchmarks for evaluating LAD, thereby establishing the foundational context for our contributions.

### 2.1 LLM-AIDED DESIGN

The integration of LLMs based on transformer architectures into hardware design is transforming the field, leveraging their proven capabilities in natural language processing to manage complex design tasks efficiently (Vaswani, 2017; Achiam et al., 2023; Touvron et al., 2023b). These models excel across various tasks by understanding and generating human-like text, which has allowed them to extend their utility to hardware design (Zheng et al., 2024; Nijkamp et al., 2022; Lozhkov et al., 2024; Lu et al., 2023). In the domain of hardware design, significant efforts focus on employing LLMs

| Name | Conference | Tests | Perturbation | Worst Coverage Score (%) | MultiModal | Difficulty tiering | Metrics |
|------|-----------|-------|--------------|--------------------------|------------|--------------------|---------|
| VeriGen (Thakur et al., 2023b) | DATE 23 | 16 modules | ✗ | – | ✗ | ✗ | Coding |
| RTLLM (Lu et al., 2024) | ASPDAC 23 | 30 designs | Partial | 52.40% | ✗ | ✗ | Coding, PPA |
| RTLLM2.0 (Liu et al., 2024a) | ICCAD24 | 50 designs | Partial | 52.40% | ✗ | ✗ | Coding, PPA |
| VerilogEval (Liu et al., 2023) | ICCAD 23 | HDLBit | Partial | 44.64% | ✗ | ✗ | Coding |
| MLLM Bench (Chang et al., 2024) | ICCAD 24 | Multimodal | ✗ | – | ✓ | ✗ | Coding |
| GenBen | This work | All criteria | ✓ | **95.17%** | ✓ | ✓ | Knowledge, Coding, Debugging, QoR |

Table 1: Comparison of Existing Work with Our Work

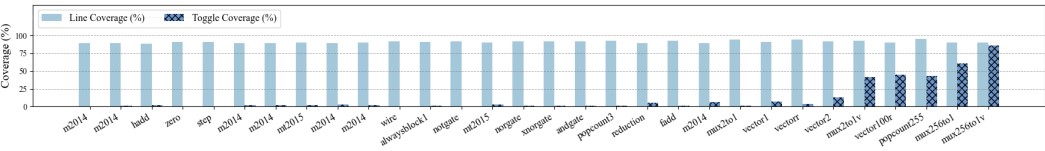

Figure 1: VerilogEval Test Coverage

to improve the generation processes and functionality of Hardware Description Languages (HDLs). Some notable projects include ChatEDA, which develops an LLM-based EDA interface that uses natural language inputs to generate task-specific code (Wu et al., 2024). The GPT4AIGChip project showcases the potential of LLM-driven design automation by modularizing various hardware functions designed specifically for AI accelerators (Fu et al., 2023). AutoChip combines LLMs with Verilog compilers to iteratively generate Verilog modules (Thakur et al., 2023c), while Chip-chat integrates conversational LLM technology to design a new 8-bit microprocessor architecture (Blocklove et al., 2023). Furthermore, ChatCPU explores a comprehensive LLM-Aided Design (LAD) chip design and introduces a novel verification methodology (Wang et al., 2024), and ChatChisel employs a specialized HDL to create a complex processor (Liu et al., 2024b). The integration of LLMs in these methods, leveraging data-based optimization techniques such as Supervised Fine-Tuning (SFT) (Hu et al., 2021; Liu et al., b; Houlsby et al., 2019; Zhang et al.; Wei et al., 2021), alongside Retrieval-Augmented Generation (RAG) (Lewis et al., 2020; Gao et al., 2023) and prompt engineering (Cao et al.; Bulat & Tzimiropoulos; Chen et al.; Deng et al.)  It is important to develop comprehensive benchmarks to mitigate the impact of pre-training and fully assess model performance in this domain.

## 2.2 BENCHMARKS FOR EVALUATING LAD

In this context, establishing benchmarks to assess the capabilities of LLMs under these adjustments is crucial (Zhong & Wang, 2023; Liu et al., a). However, existing benchmarks are static and open-source, making them susceptible to unintentional inclusion in pre-training datasets, and there is room for improvement in testbench coverage, benchmark data diversity, and the scalability of evaluation metrics. For instance, although Verigen (Thakur et al., 2023a) evaluated 17 designs after fine-tuning CodeGen (Nijkamp et al., 2022), the assessments mainly targeted simple and small-scale circuit designs, and these benchmarks are not open source. RTLLM (Lu et al., 2024) and RTLLM2.0 (Liu et al., 2024a) provided 30-50 testbenches for testing LLMs. These testbenches were evaluated using VCS to determine verification coverage, with the worst coverage score being approximately 52.40%, as shown in Table 1. Additionally, the testbenches featured relatively simple and uniform question types, and some of the mentioned evaluation tools are not open-source. VerilogEval (Liu et al., 2023) introduced a comprehensive dataset of 156 problems from HDLBits for automated functional correctness testing of LLM-generated Verilog code. However, these benchmarks are relatively easy, and models that perform best have high verification pass rates, which do not allow for further stress testing as models continue to evolve. In addition, the worst verification coverage of VerilogEval is relatively low at 44.63%. In order to investigate the test coverage limitation, we further analyze the VerilogEval benchmark. As shown in Figure 1. RTL-Repo (Allam & Shalan, 2024), while assessing the RTL Repo project, can evaluate LLM accuracy through exact matching (EM) and edit similarity (ES), yet such metrics do not guarantee that the LLM-generated designs are verifiable or optimally synthesizable. PyHDL-Eval (Batten et al., 2024) and VHDLEval (Vijayaraghavan et al., 2024) are domain-specific benchmarks whose data diversity and evaluation metrics could be further enriched. HDLEval (Zakharov & Renau) initiated a multifunctional benchmark that uses rapid engineering techniques to overcome syntactical differences across HDLs and adopts formal verification methods to assess code generated across multiple HDLs. However, there is still room to enhance testbench

coverage and the richness of question types. ChipGPTV (Chang et al., 2024) proposed using visual representations to clarify design intentions and introduced a tiered benchmark to assess MLLM performance in Verilog generation, but there is still further scope to expand the diversity of code generation and hardware design knowledge testing metrics. A detailed comparison of existing work with our work can be found in Table 1.

## 2.3 PROBLEM FORMULATION

- **1. Verification Coverage Gaps**: Existing benchmarks reveal a gap in design complexity and verification coverage. The developed testbenches often fail to adequately represent the essential function points of the included RTL designs, a situation that worsens as design complexity increases. Consequently, the limited verification coverage of generated hardware can undermine the authenticity of evaluation results.

- **2. Deficient Data Diversity**: Current benchmark problems demonstrate insufficient diversity and richness in data sources and modalities. Many benchmarks sourced from educational materials are overly simplistic and lack silicon validation. Furthermore, these text-based, unimodal benchmarks often fail to reflect real-world design specifications, which frequently incorporate visual schematics and timing diagrams.

- **3. Benchmark Test Set Contamination:** Since these benchmarks are statically open-source on GitHub, associated RTL designs and specifications can be automatically captured by crawlers as part of the RTL language datasets. Evolving LLMs like GPT-4, Claude, and Llama 3 may inadvertently incorporate this data during pre-training, resulting in data leakage and contamination of the test set.

- **4. Limited Evaluation Metrics:** Existing benchmarks focus primarily on syntax and functional pass rates, neglecting critical QoR metrics such as PPA statistics and synthesizability. This oversight can lead to an incomplete evaluation of the generated designs.

## 3 DESIGN & PHILOSOPHY

In this section, we introduce the detailed GenBen design including workflow, dataset collection, task construction, data perturbation, quality enhancement, and question generation.

### 3.1 DESIGN STRATEGIES OF GENBEN

Targeting the challenges in Section 2.3, the GenBen design incorporates the following strategies:

- **Improved Dataset Diversity:** Curated from sources like GitHub, silicon-proven projects, and StackOverflow, featuring objective (knowledge) and subjective (coding, debugging, design optimization) tests, categorized into three difficulty levels (Table 2).

- **Coverage-Enhanced TestBench:** The quaility of testbench are enhanced in line, toggle, and functional coverage by our experts to ensure fine-grained verification.

- **Perturbed Generative Benchmark:** Employs perturbation strategies during test generation and evaluation to defend against memorization.

- **Multi-Dimensional Evaluation:** Design five dimensions and 12 sub-items featuring QoR aware mechanism as shown in (Table 5), enabling flexible, custom benchmarks.

### 3.2 GENBEN FRAMEWORK & WORKFLOW

The GenBen framework has below key components: a pre-processed test set, a task generator, a dynamic perturbator, a response collector, an evaluation suite, a report analyzer, and a scoring module.

Evaluation begins with the user providing the API of the model and modality information as shown in Figure 2.B. GenBen then generates test tests from the test dataset $\mathcal{D}$ using scripts, denoted as $\mathcal{T}$ which remain consistent for each evaluation tests. Subsequently, the dynamic perturbation component applies surface-level perturbations to $\mathcal{T}$, resulting in a transformed set $\mathcal{T}'$. These perturbations introduce slight variations for dynamic evaluation. GenBen collects responses from the model for

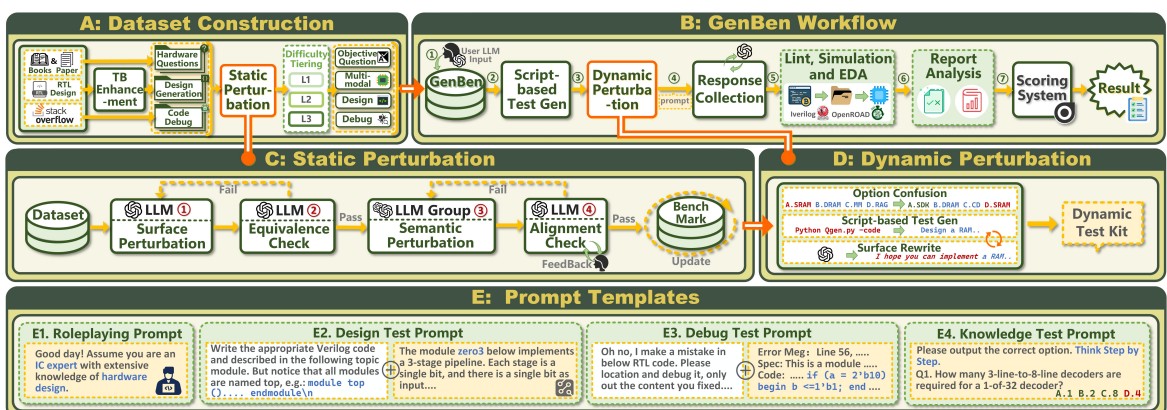

Figure 2: GenBen Pipeline

Table 2: Difficulty Tiering

| Categories | Description |
|---|---|
| L1 (Simple) | Suitable for initial evaluation, focusing on fundamental concepts and straightforward tests.. |
| L2 (Intermediate) | Involving more complex tests and requiring robust problem-solving skills. |
| L3 (Tough) | Tackling real-world design challenges and requiring advanced reasoning & implementation capabilities. |

both $\mathcal{T}$ and $\mathcal{T}'$ using a unified prompt template. These responses are then fed into the evaluation suite, which performs checks and executions to validate the outputs. GenBen simulates the generated answers and corresponding testbenches using Icarus Verilog (Iverilog) to obtain reports on syntax and functional correctness. Designs that pass the functional tests undergo further physical implementation using the open-source SkyWater 130nm Process Design Kit (PDK)(sky, 2020) and the OpenLane flow. Within OpenLane, the Yosys(Wolf et al., 2013) component extracts data on synthesizability, area, and power, while OpenSTA(Cherry, 2023) handles timing-related data extraction. The report analyzer then extracts metric-related information from the evaluation results. This information is passed to the scoring module, which evaluates the performance of the model based on predefined metrics and generates the final results.

## 3.3 BENCHMARK DATASET CONSTRUCTION

Our dataset construction process is illustrated in Figure 2.A. We collected hardware-related content from across the web, which was then meticulously curated by a team of 10 domain experts. These experts screened the data for correctness, completeness, and diversity, with a particular focus on sampling from silicon-proven projects. For selected code tests, we enhanced their testbenches to ensure robust evaluation as shown in Section 3.3.1; for debug test, we refined them as shown in Seciton 3.3.2.

The collected and refined content was then filtered and categorized into three types of tests: knowledge, design, and debugging. To mitigate the interference of publicly available pre-training data on the evaluation, we introduced static perturbations. Using a multi-agent system combined with human feedback as shown in Figure 2.C, we applied perturbations to the tests, transforming them into new content at the token sequence level.

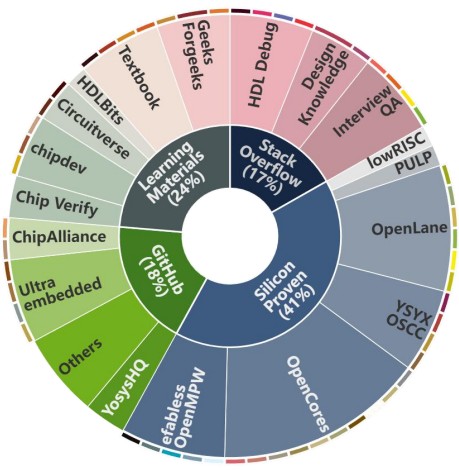

Figure 3: Dataset of GenBen

Table 3: Test Categories in GenBen

| Test | Amount | Description |
|------|--------|-------------|
| Knowledge Master | 75 | Focus on evaluating the grasp of the LLM on fundamental hardware concepts and principles. |
| Knowledge Transfer | 69 | Apply concepts to new and complex scenarios for generalization. |
| Design | 99 | Divide the difficulty based on the number of lines of code, type,and design time. |
| Debug | 57 | Distinguish the difficulty of correcting syntax/function/combination errors. |
| Multimodal | 60 | Incorporate both textual and visual inputs. |

The updated tests were then tiered according to difficulty, as shown in Table 2, and mapped to different categories of tests: objective tests (assessing basic knowledge understanding and transfer), design tests, debugging tests, and multimodal tests. This mapping ensures comprehensive end-to-end evaluation of the knowledge and capabilities of the LLM.

Ultimately,the GenBen tests are shown in Table 3 with distribution across difficulty levels.

### 3.3.1 TESTBENCH COVERAGE ENHANCEMENT

Following the preparation of the GenBen datasets, we proceed to build testbenches for each RTL design to enhance the verification coverage of generative designs. We rigorously employ a standard, end-to-end verification flow that ensures a point-to-point mapping between the generated stimuli and the functional coverage checklist. By employing constraint randomization and coverage-driven testbench generation methodologies, we significantly improve the verification coverage for each generated RTL design, thereby maximizing the efficacy of benchmarking LAD capabilities.

### 3.3.2 DEBUG TEST DESIGN

Moreover, the debugging process is a critical step in the integrated circuit design flow and should not be omitted from benchmarking: real-world hardware design often involves identifying and correcting errors. Therefore, we introduce debugging tests in GenBen. We categorize them into three types: *syntax errors*, *functional errors*, and *a hybrid of both*. By injecting errors into correct designs, we create debugging datasets that require LLMs to locate and fix the erroneous code.

### 3.4 DATA PERTURBATION

Building upon insights from existing DS-1000 works (Lai et al., 2023), we introduced a perturbation strategy to mitigate potential memorization biases in AI models. We implemented two types of perturbations: surface and semantic as shown in Table 4. Surface-level perturbations alter the phrasing of a question without changing its core meaning. For instance, the prompt "*Design a 128x32 RAM module*" might be rephrased as "*Construct a memory module with 128 addresses and 32-bit data width*". As illustrated in Figure 2.C, surface perturbations require a equivalence check to ensure that the meaning of the task remains unchanged.

Table 4: Perturbation Categories

| Perturbation | Description |
|--------------|-------------|
| Surface | Paraphrase: don't change reference solution |
| Semantic | Generalization: will change reference solution |

Semantic perturbations increase the difficulty of a task by altering its underlying meaning. For example, changing a prompt from "*Design a 16-bit adder*" to "*Design an adder that can handle arithmetic of two complements for 16-bit inputs*" requires the model to exhibit stronger reasoning abilities. It is necessary to align the updated tasks with their corresponding solutions to maintain consistency as shown in Figure 2.C.

We implemented perturbations in two stages: during the construction of GenBen, as shown in Figure 2.A, and throughout the GenBen workflow, as depicted in Figure 2.B.

### 3.4.1 STATIC PERTURBATION

Static perturbations are applied during the test construction phase, leveraging the multi-agent process illustrated in Figure 2.C. This process involves adding surface and semantic perturbations to candidate tests, which are then reviewed by human experts to finalize the test design. Key aspects of this stage include: 1).Abstracting concepts, definitions, and computational problems into objective questions; 2).Injecting bugs into correct code to create debugging tests; and 3).Adjusting and deriving new coding tests. These perturbations are applied at the data source level and remain unchanged once the test set is finalized.

### 3.4.2 DYNAMIC PERTURBATION

To further reduce the interference of pre-training data, we introduce dynamic perturbations during the evaluation process using surface-level perturbations. This stage involves generating slightly varied versions of the tests as described in Section 3.2. This provides researchers with additional insights and references for analyzing the robustness and adaptability of the LLMs.

## 3.5 MULTIMODAL FEATURE SUPPORT

The GenBen framework offers both unimodal and multimodal task evaluations, addressing the growing need for comprehensive assessment methodologies in hardware design. This feature is particularly important because real-world design processes often require the integration of various forms of data, such as textual specifications, diagrams, and architectural schematics. Understanding and synthesizing information from multiple modalities is crucial for effective hardware design.

In GenBen, multimodal data types include basic circuit diagrams, design architecture schematics, waveform diagrams, and tables. These data types are utilized across various test categories: knowledge questions assess the understanding of fundamental concepts and their applications; code generation tests require interpreting and translating visual schematics into HDL code; and debugging tests involve identifying and correcting errors in designs that are presented through a combination of text and visual data.

## 3.6 EVALUATION METRIC DESIGN

We developed a comprehensive evaluation metric system, as detailed in Table 5, which includes both basic correctness metrics and QoR metrics. The QoR metrics—encompassing synthesizability, power, area, and timing performance for evaluating the feasibility of generated designs for silicon implementation. To quantify the design optimization capability of LLMs, we normalize these QoR results against a reference design for result-aware.

This comprehensive approach, which includes knowledge master & transfer, design generation, debugging, multimodal content and design optimization derived from post-synthesis, enables GenBen to systematically evaluate LLM performance throughout the entire hardware design process. Especially, the improvement-aware metrics derived from power, area, and timing analyses offer a clear and intuitive representation

Table 5: Metrics of GenBen

| Metric | Description |
|---|---|
| Knowledge Master | Basic concept without need of deduction |
| Knowledge Transfer | Generalization skills that need CoT or deduction |
| Debug Ability | Skills in issue-solving and perseverance |
| Code Correctness | **Syntax** & **Function:** Skills in programming |
| Quality of Result | **Synthesizability**, **Power**, **Area & Timing** |

of the capability of the model to produce high-quality, manufacturable hardware designs.

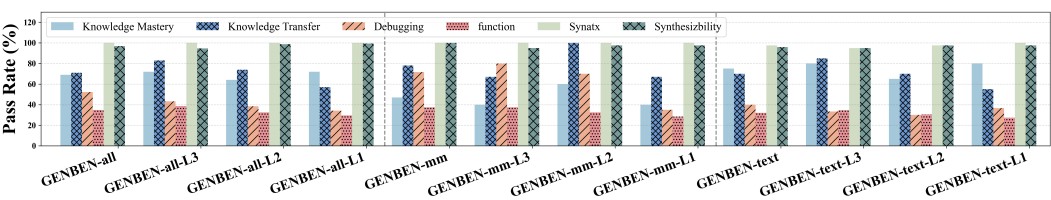

Figure 4: GPT-4o Tests

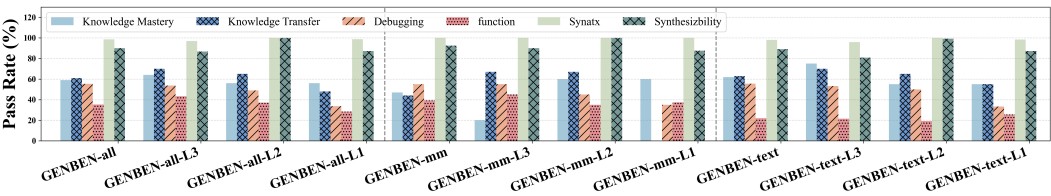

Figure 5: Claude3.5 Tests

## 4 EXPERIMENTAL RESULTS

### 4.1 EXPERIMENTAL SETUP

**Model Selection:** Our study evaluated nine models, comprising six multimodal and three language models. The selected models are GPT-4-turbo, GPT-4o, GPT-3.5-turbo, Claude3.5, Llama3, QWEN-vl-max, QWEN-vl-plus, GLM-4V-plus, and GLM-4.

**Prompt Template:** We developed a standardized prompt structure consisting of two key components: (1) a role-playing prompt and (2) a problem description prompt as shown in Figure 2.E.

**Test Iteration:** We employed a pass@5 evaluation strategy throughout our experiments.

**Pass Rate:** Finallys, we used Pass Rate (PR) to quantify the overall ability. For an problem $\theta_i$ and its LLM-generated answer $\theta_i^*$, we had a corresponding set of correct answer in GenBen database $(x_i^0, y_i^0), (x_i^1, y_i^1), \ldots, (x_i^m, y_i^m)$. For the correct solution, $\theta_i^*$, it should produce the correct output $y_i^j$ when applied to the input data $x_i^j$ from the test cases. That is, $a_{\theta_i^*}\left(x_i^j\right) = y_i^j$, the test case $\left(x_i^j, y_i^j\right)$ can be regarded as passing. Whether the answer is successfully passed can be described as $\bigwedge_{j=0}^{m}\left[a_{\theta_i^*}\left(x_i^j\right) = y_i^j\right]$, an aggregate result of all test cases. The PR are defined as:

$$\mathbf{PR} = \sum_{i=0}^{n} \frac{\bigwedge_{j=0}^{m}\left[a_{\theta_i^*}\left(x_i^j\right) = y_i^j\right]}{n} \times 100\% \tag{1}$$

**Evaluation Criteria:**

- **Knowledge& debugging tests.** Pass/fail criterion, comparing with reference.
- **Code generation.** *Syntax*: failed attempts receive a score of 0%. Successful attempts with warnings incur a 5% penalty per warning, with a minimum score of 60%. *Function*: calculated ranging from 0% to 100%. Besides, to assess QoR optimization capabilities, we conduct a normalized comparison against a reference design.

### 4.2 RESULTS ANALYSIS

**Stable Benchmark Performance:** Results shown in Figure 4-12 highlight that the best model achieved a overall PR slightly above 40% but below 50%, aligning with expectations.

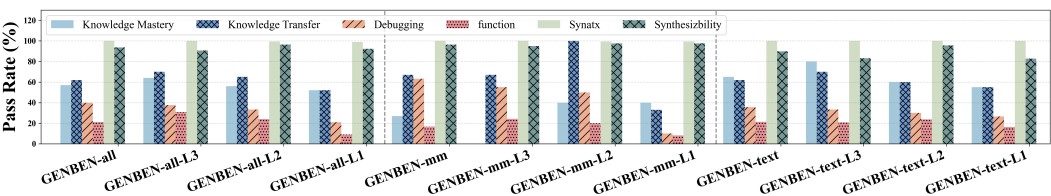

Figure 6: GPT-4-turbo Tests

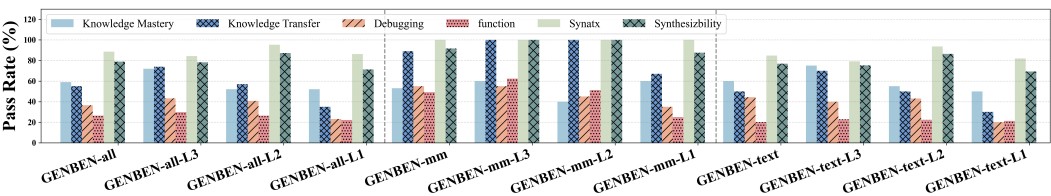

Figure 7: QWEN-vl-max Tests

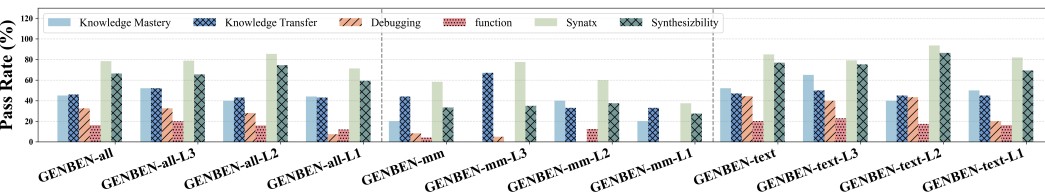

Figure 8: QWEN-vl-plus Tests

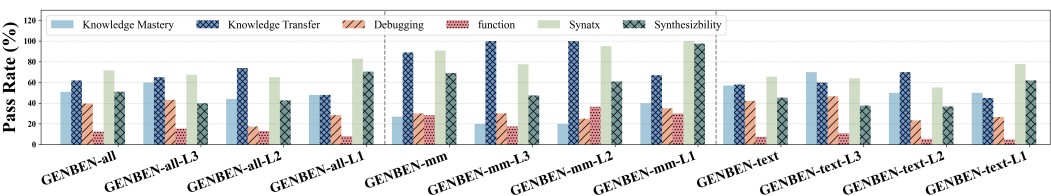

Figure 9: GLM-4V-plus Tests

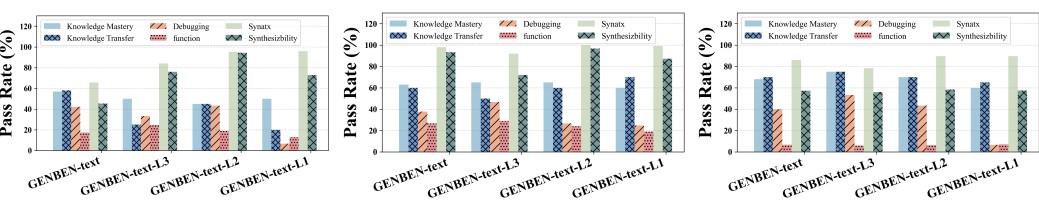

Figure 10: GLM-4 Tests    Figure 11: GPT-3.5-turbo Tests    Figure 12: Llama3 Tests

**Effective Difficulty Tiering:** Difficulty levels and PRs have a correlation. Using GPT-4o (shown in Figure 4, detailed value in Section A, Table 10) as a example, the consistent 5-10% difference in PRs across these levels.

**Correlation Between Tests:** The data indicates a correlation between Knowledge Mastery and coding abilities. Models that performed well in Knowledge Mastery, such as GPT-4o and Claude 3.5, also showed high scores in Debugging and Functional Correctness. This suggests that a solid understanding of fundamental concepts positively influences practical coding skills.

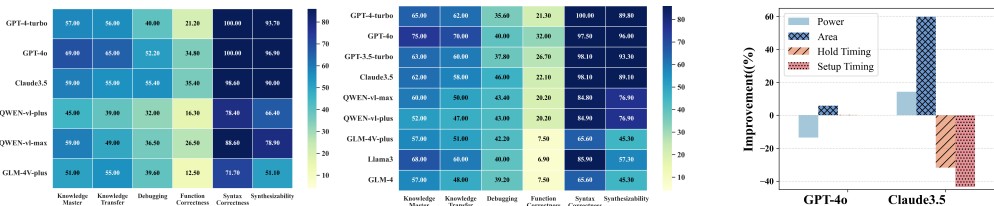

Figure 13: PR of All Tests          Figure 14: PR of Text Tests          Figure 15: Example of QoR

**Synthesizability vs. Syntax Discrepancy:** Synthesizability and syntax correctness has a high inconsistency (**91.76%**), as Figure 13 and 14 shown. This discrepancy arises from the inherent differences in requirements between simulation and synthesis tools, exacerbated by the presence of non-IEEE-compliant code in pre-training datasets. This issue highlights an area for future model improvement.

**Debugging Capabilities:** Models generally exhibit stronger debugging capabilities compared to code generation, which may be attributed to the additional context provided in debugging tests.

**QoR Analysis for Top Models** The QoR result for GPT-4o and Claude 3.5 is presented in Figure 15. GPT-4o shows stable performance across area and timing metrics with improvement need in low-power design. On the other hand, Claude3.5 demonstrates aggressive optimization in power and area but at the cost of timing violations. These insights shows the different trade-offs by different models.



Figure 16: Example of DP Influence

**Ablation Experiment of Dynamic Perturbation** Figure 16 takes Llama3 as an example to illustrate the impact of dynamic perturbations from GPT-3.5 and GPT-4. The results demonstrate that the performance fluctuated across different test sets, with an overall performance decline of approximately 9%.

## 5    CONCLUSION

In this paper, we introduce GenBen, a comprehensive benchmark designed to evaluate the capabilities of LLMs in the domain of hardware design. Unlike existing benchmarks that primarily focus on code generation, GenBen offers a more holistic evaluation by encompassing debugging, optimization, and the chip hardening flow. By introducing perturbations and hierarchical task classification, GenBen provides a diverse range of end-to-end, open-source evaluation modalities. Our goal is to establish GenBen as a catalyst for advancements in LAD, providing a reliable benchmark for generative hardware designs tailored to meet real-world silicon manufacturing requirements.

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

# A APPENDIX

## A.1 CONCEPT OF LLM-AIDED DESIGN

*LLM-Aided Design* (LAD) is defined as the use of *Large Language Models* (LLMs) as a methodology to assist in designing circuits, software, and computing systems with improved quality, productivity, robustness, and cost-effectiveness. It focuses on discussing results that leverage the significant advancements and innovations captured by generative AI and LLM technology to offer new methods and solutions for design automation targeting various applications. This concept was first introduced by IEEE ICCAD 2023.

## A.2 QUALITY OF RESULTS IN HARDWARE DESIGN

In hardware design, *Quality of Results* (QoR) metrics are crucial for evaluating the effectiveness and efficiency of a design. These metrics encompass various aspects that determine the practicality and performance of the generated hardware. Below, we provide detailed explanations of key QoR metrics and their significance:

### A.2.1 SYNTHESIZABILITY

*Synthesizability* refers to the ability of a hardware design to be translated from a high-level description into a gate-level netlist that can be fabricated. This process, known as *synthesis*, is fundamental to the hardware design flow. A design that is not synthesizable cannot be implemented in silicon, rendering it impractical for real-world applications. Ensuring synthesizability is the first step in verifying that a design can transition from concept to physical implementation. It is important to note that a design passing simulation does not guarantee it will pass synthesis, often due to syntax or structural issues that, while acceptable in simulation, do not meet the stringent requirements of synthesis tools.

### A.2.2 POWER, PERFORMANCE, AND AREA (PPA)

*Power, Performance, and Area* (PPA) is a comprehensive set of metrics used to evaluate the efficiency of a hardware design:

- **Power**: Measures the amount of electrical power consumed by the hardware design. Lower power consumption is critical for battery-operated devices and energy-efficient systems.

- **Performance**: Often evaluated in terms of maximum operating frequency or throughput, performance metrics indicate how fast the hardware can operate. Higher performance is essential for applications requiring rapid data processing and high-speed computations.

- **Area**: Refers to the silicon area occupied by the hardware design. Minimizing area is important for reducing manufacturing costs and enabling the integration of more functionality within a given chip size.

Balancing these three aspects—power, performance, and area—is a key challenge in hardware design, as improvements in one area often lead to trade-offs in the others.

In our benchmark design, to ensure consistency and efficiency in runtime and EDA script standardization, we have unified the primary performance metric to *frequency*. Consequently, performance feedback is primarily provided through *Total Negative Slack* (TNS) and *Worst Negative Slack* (WNS).

### A.2.3 TOTAL NEGATIVE SLACK (TNS) AND WORST NEGATIVE SLACK (WNS)

*Total Negative Slack* (TNS) and *Worst Negative Slack* (WNS) are critical timing metrics used to evaluate the timing performance of a hardware design:

- **Total Negative Slack (TNS)**: The sum of all negative timing slacks in a design. Negative slack indicates that a timing path does not meet its required timing constraints. TNS provides an aggregate measure of timing violations across the entire design.

- **Worst Negative Slack (WNS)**: Represents the most severe timing violation in the design. It is the largest single negative slack value and highlights the worst-performing timing path.

Both TNS and WNS are essential for identifying and addressing timing issues, ensuring that the design meets its performance requirements without violations.

### A.2.4 SETUP AND HOLD TIMES

*Setup* and *hold times* are critical parameters for ensuring reliable operation of sequential circuits:

- **Setup Time**: The minimum time before the clock edge by which data must be stable to be correctly latched. Violations in setup time can lead to incorrect data being captured, affecting the functionality of the design.

- **Hold Time**: The minimum time after the clock edge during which data must remain stable to be correctly latched. Violations in hold time can cause data corruption, leading to unpredictable circuit behavior.

Ensuring that setup and hold times are met is crucial for the stability and reliability of the hardware design.

In summary, these QoR metrics provide a comprehensive framework for evaluating the practical viability and performance of hardware designs. They are essential for ensuring that a design not only meets its functional requirements but also operates efficiently and reliably in real-world applications. Moreover, addressing the syntactical and structural requirements for synthesis ensures that designs are theoretically sound and practically implementable in silicon.

### A.3 THE ROLE OF OPEN-SOURCE EDA TOOLS IN ENHANCING SCIENTIFIC REPRODUCIBILITY

Open-source *Electronic Design Automation* (EDA) tools are key enablers of scientific reproducibility, providing accessible alternatives to benchmarks that have traditionally relied on commercial EDA tools such as *Design Compiler* and *Synopsys VCS*.

One of the primary advantages of open-source EDA tools is their facilitation of effortless collaboration among researchers and designers. They eliminate the need for complex legal agreements such as *Non-Disclosure Agreements* (NDAs), allowing for straightforward sharing of designs, ideas, and

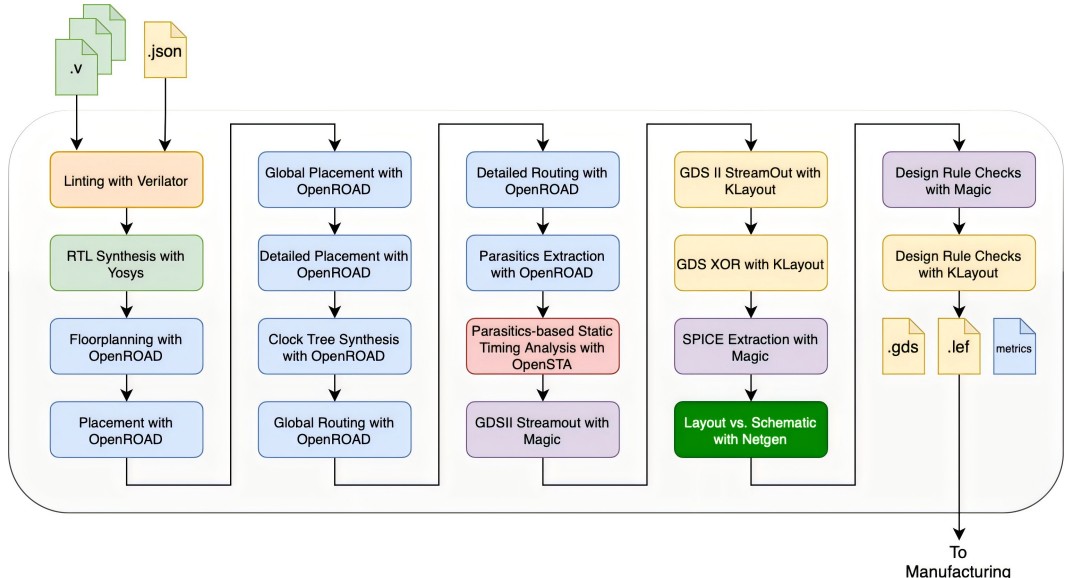

Figure 17: OpenLane Flow

materials. This ease of collaboration is particularly beneficial for integrating experts from fields like computer science, where open-source development is prevalent.

Moreover, open-source EDA tools are invaluable for educational and research purposes. They enable educators to provide students with practical insights into the design automation process. Students and researchers can modify the code, test their hypotheses, and gain a comprehensive understanding of the chip design process.

### A.3.1 IMPLEMENTATION OF OPEN-SOURCE EDA TOOLS IN GENBEN

In our *GenBen* design process, we exclusively use open-source EDA tools. During the task construction phase, we rely on *Verilator* to perform coverage analysis, enhancement, and refinement of the testbenches. For agile execution during model testing, we use *Icarus Verilog* due to its faster compilation times, although it lacks comprehensive coverage analysis. Therefore, we employ different tools at various stages to balance efficiency and thoroughness.

Additionally, to obtain physical implementation information, we use *OpenLane*, an open-source RTL-to-GDSII EDA flow, as illustrated in Figure 17. OpenLane enables us to extract critical data on synthesizability, area, power, and timing, ensuring that our benchmarks are both practical and reproducible using widely accessible tools.

### A.3.2 CHOICE OF PDK FOR QOR EVALUATION

The *Quality of Results* (QoR) of a design can vary significantly across different *Process Design Kits* (PDKs). To ensure consistency in our evaluations, we have chosen the open-source *SkyWater 130nm* PDK for QoR testing. This choice provides a standardized reference point for assessing the practical viability of hardware designs, allowing for fair and comparable results across different design implementations.

### A.4 SOURCES OF OUR DATASET

The dataset for our *GenBen* benchmark is meticulously curated from a diverse array of sources to ensure comprehensive coverage of various aspects of hardware design. These sources are categorized into three levels—*Level 1* (L1), *Level 2* (L2), and *Level 3* (L3)—based on the complexity and depth of the tasks they contribute.

**Level 1 (L1)** sources provide fundamental tasks aimed at assessing basic knowledge and skills in hardware design. These include materials such as university textbooks, which supply essential theoretical and practical questions for understanding core concepts. Basic code examples offer simple coding tasks to test foundational programming skills, while basic quizzes include multiple-choice and short-answer questions to evaluate basic knowledge. Additionally, *HDLBits* provides elementary hardware description language (HDL) exercises suitable for beginners.

**Level 2 (L2)** sources present intermediate-level tasks that require a deeper understanding and application of hardware design principles. These sources incorporate *GitHub* projects that provide real-world coding examples and projects necessitating practical implementation skills. Graduate projects contribute tasks from advanced coursework, focusing on more complex design and problem-solving abilities. Question and answer forums such as *Stack Overflow* and *GitHub Q&A* include practical debugging and problem-solving questions commonly encountered by developers, addressing real-world issues faced by practitioners.

**Level 3 (L3)** sources deliver advanced tasks that challenge the highest level of expertise in hardware design. These include silicon-proven repositories, contributing tasks from projects successfully implemented in silicon, ensuring high reliability and complexity. Research textbooks provide advanced theoretical and practical problems stemming from cutting-edge research in hardware design. Peer-reviewed publications from *ACM* and *IEEE* include tasks based on recent advancements in the field. Student contests offer challenging problems from hardware design competitions, while studies in advanced microarchitecture supply tasks involving sophisticated architectural design and optimization. Innovative projects introduce problems that push the boundaries of current technology, and industrial projects provide tasks derived from real-world industrial applications, emphasizing practical implementation and optimization.

The tasks from these varied sources are further categorized to cover a wide range of skills and knowledge areas. Tasks focused on *knowledge transfer* assess the ability to apply learned concepts to new scenarios, enhancing adaptability in design approaches. Those involving *code debugging* require identifying and correcting errors in code, which is critical for developing robust hardware systems. *Knowledge mastery* tasks evaluate the depth of understanding of fundamental concepts, ensuring a solid theoretical foundation. *Code generation* tasks necessitate the creation of new code based on given specifications, testing the ability to innovate and implement design requirements effectively.

These tasks are organized into two main categories for the GenBen benchmark: *text-based* tasks and *multimodal* tasks. Text-based tasks are purely textual, focusing on theoretical and conceptual understanding, including problem-solving and analytical reasoning. Multimodal tasks involve multiple forms of data, such as text and diagrams, to simulate real-world design challenges and provide a more comprehensive assessment of practical skills.

Figure 20 illustrates the relationship between the data sources and the final dataset. Notably, a significant portion of silicon-proven designs comes from resources such as Google FOSS and OpenCores, as shown in Figures 18 and 19.

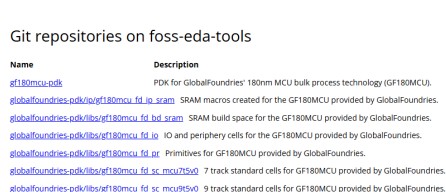

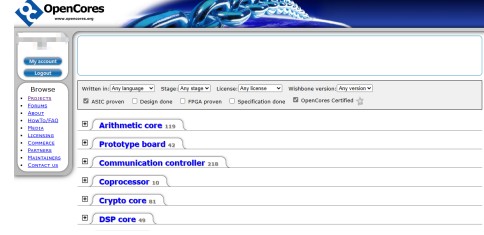

Figure 18: FOSS Projects of OpenMPW      Figure 19: OpenCores

### A.5 GENERATIVE BENCHMARK CONCEPT AND PRINCIPLES

The concept of a *generative benchmark* involves creating evaluation tasks that are not directly stored in plaintext on platforms like GitHub but are instead implicitly distributed across various datasets. This approach requires the use of scripts to dynamically extract tasks, arrange options, and ran-

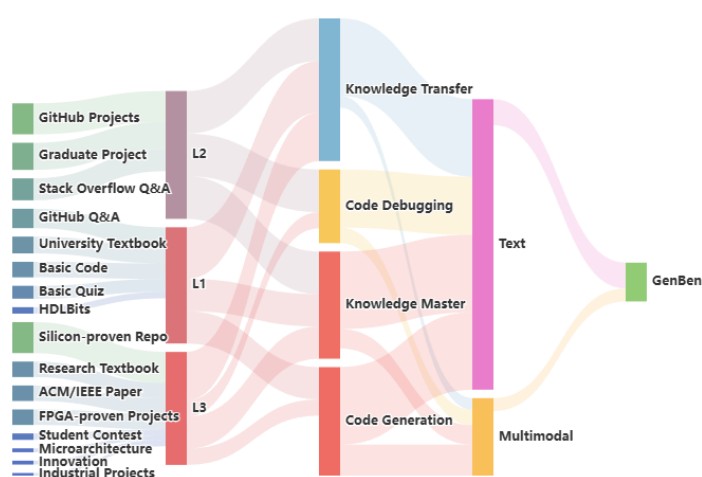

Figure 20: Data Sources of the GenBen Dataset

domize the order of questions each time they are generated. Such a methodology helps mitigate the interference caused by a model's pre-training memory, ensuring that assessments are based on competency rather than memorization.

The principle behind this generative approach is to ensure that each generated task remains consistent for every evaluation, thereby maintaining the objectivity and fairness of the assessments. Additionally, a control group with only surface-level perturbations is introduced, allowing for simultaneous evaluation of both groups and providing insights into the model's sensitivity to such variations.

Moreover, GenBen supports researchers in replacing or modifying the evaluation methods and tasks, as the tests, evaluation framework, and generative scripts are decoupled. This flexibility allows for the adaptation of the benchmark to different research needs and the incorporation of new evaluation strategies. Below are the test generation algorithm 1 and the evaluation flow 2, which detail the processes involved in generating and assessing the benchmark tasks.

### A.5.1 TEST GENERATION ALGORITHM

---

**Algorithm 1** Test Generation Algorithm

---

**Require:** Test dataset $\mathcal{D}$
**Ensure:** Generated test set $\mathcal{T}$ and perturbed test set $\mathcal{T}'$
 1: Initialize test set $\mathcal{T} \leftarrow \emptyset$
 2: Initialize perturbed test set $\mathcal{T}' \leftarrow \emptyset$
 3: Load test dataset $\mathcal{D}$
 4: **for** each test $d \in \mathcal{D}$ **do**
 5:     Generate task $t$ from $d$ using script
 6:     Add task $t$ to $\mathcal{T}$
 7: **end for**
 8: **for** each task $t \in \mathcal{T}$ **do**
 9:     Apply surface-level perturbation to $t$ to generate $t'$
10:     Add perturbed task $t'$ to $\mathcal{T}'$
11: **end for**
12:
13: **return** $\mathcal{T}$ and $\mathcal{T}'$

---

Table 6: Results of Tested Multimodal Models on GenBen-all

| | model | Knowledge Master | Knowledge Transfer | Debugging | Function Correctness | Synatx Correctness | Synthesizbility |
|---|---|---|---|---|---|---|---|
| GENBEN-all | gpt-4-turbo | 57.00% | 56.00% | 40.00% | 21.20% | 100.00% | 93.70% |
| GENBEN-all | gpt-4o | 69.00% | 65.00% | 52.20% | 34.80% | 100.00% | 96.90% |
| GENBEN-all | claude3.5 | 59.00% | 55.00% | 55.40% | 35.40% | 98.60% | 90.00% |
| GENBEN-all | qwen-vl-plus | 45.00% | 39.00% | 32.00% | 16.30% | 78.40% | 66.40% |
| GENBEN-all | qwen-vl-max | 59.00% | 49.00% | 36.50% | 26.50% | 88.60% | 78.90% |
| GENBEN-all | GLM-4V-plus | 51.00% | 55.00% | 39.60% | 12.50% | 71.70% | 51.10% |

Table 7: Results of All Tested Models on GenBen-Text

| | model | Knowledge Master | Knowledge Transfer | Debugging | Function Correctness | Synatx Correctness | Synthesizbility |
|---|---|---|---|---|---|---|---|
| GENBEN-text | gpt-4-turbo | 65.00% | 62.00% | 35.60% | 21.30% | 100.00% | 89.80% |
| GENBEN-text | gpt-4o | 75.00% | 70.00% | 40.00% | 32.00% | 97.50% | 96.00% |
| GENBEN-text | gpt-3.5-turbo | 63.00% | 60.00% | 37.80% | 26.70% | 98.10% | 93.30% |
| GENBEN-text | claude3.5 | 62.00% | 58.00% | 46.00% | 22.10% | 98.10% | 89.10% |
| GENBEN-text | qwen-vl-max | 60.00% | 50.00% | 43.40% | 20.20% | 84.80% | 76.90% |
| GENBEN-text | qwen-vl-plus | 52.00% | 47.00% | 43.00% | 20.20% | 84.90% | 76.90% |
| GENBEN-text | GLM-4V-plus | 57.00% | 51.00% | 42.20% | 7.50% | 65.60% | 45.30% |
| GENBEN-text | llama3 | 68.00% | 60.00% | 40.00% | 6.90% | 85.90% | 57.30% |
| GENBEN-text | GLM-4V-plus | 57.00% | 48.00% | 39.20% | 7.50% | 65.60% | 45.30% |

---

**Algorithm 2** Total Evaluation Flow

---

**Require:** Test set $\mathcal{T}$, Perturbed test set $\mathcal{T}'$, Model's API $\mathcal{A}$, Modality information $\mathcal{M}$
**Ensure:** Evaluation results and final scores
 1: Initialize response set $\mathcal{R} \leftarrow \emptyset$
 2: Initialize perturbed response set $\mathcal{R}' \leftarrow \emptyset$
 3: Initialize evaluation results $\mathcal{E} \leftarrow \emptyset$
 4: Initialize final scores $\mathcal{S} \leftarrow \emptyset$
 5: **for** each task $t \in \mathcal{T}$ **do**
 6:     Collect response $r$ from model using $\mathcal{A}$
 7:     Add response $r$ to $\mathcal{R}$
 8: **end for**
 9: **for** each perturbed task $t' \in \mathcal{T}'$ **do**
10:     Collect response $r'$ from model using $\mathcal{A}$
11:     Add response $r'$ to $\mathcal{R}'$
12: **end for**
13: **for** each response $r \in \mathcal{R}$ and $r' \in \mathcal{R}'$ **do**
14:     Validate $r$ and $r'$ using evaluation suite
15:     Simulate $r$ and $r'$ with Iverilog
16:     Generate syntax and functional correctness reports
17:     **if** $r$ and $r'$ pass functional tests **then**
18:         Perform physical implementation using SkyWater 130nm PDK and OpenLane
19:         Extract synthesizability, area, and power data with Yosys
20:         Extract timing-related data with OpenSTA
21:     **end if**
22:     Add evaluation results to $\mathcal{E}$
23: **end for**
24: Analyze evaluation results in $\mathcal{E}$ using report analyzer
25: Generate final scores $\mathcal{S}$ based on predefined metrics
26:
27: **return** $\mathcal{S}$

---

A.6  EXPERIMENTAL RESULTS

We categorized the tasks into three groups: **GenBen-all**, **GenBen-mm**, and **GenBen-text**, corresponding to all tasks, multimodal tasks, and text-based tasks, respectively. Additionally, the latter two categories are further classified into levels **L1** to **L3**.

Table 6 shows the results of tested multimodal models on all tests and Table 7 shows the results of all models on unimodal tests. Table 8 and 9 respectively present the PPA data of the Claude 3.5 and GPT-4 models for QoR analysis.

Table 8: PPA Info of Claude3.5 on Part of Generated Design

| Modal | Function Correctness | Area | | Power | | Hold WNS | | Setup TNS | |
|---|---|---|---|---|---|---|---|---|---|
| | | Generated | Reference | Generated | Reference | Generated | Reference | Generated | Reference |
| Text | 0.4 | 6.256 | 3.7536 | 6.33E-07 | 5.92E-07 | 3.8839 | 3.8395 | 5.5943 | 5.6193 |
| | 0.4 | 7.5072 | 5.0048 | 7.01E-07 | 6.85E-07 | 3.9746 | 3.9153 | 5.504 | 5.5586 |
| | 0.2 | 6.256 | 6.256 | 6.93E-07 | 6.93E-07 | 3.9485 | 3.9485 | 5.3708 | 5.3708 |
| | 1 | 22.5216 | 22.5216 | 1.63E-06 | 1.63E-06 | 3.8877 | 3.8877 | 5.317 | 5.317 |
| | 0.8 | 22.5216 | 22.5216 | 1.63E-06 | 1.63E-06 | 3.8877 | 3.8877 | 5.317 | 5.317 |
| | 0.2 | 73.8208 | 73.8208 | 1.35E-05 | 1.35E-05 | 0.1141 | 0.1141 | 6.9101 | 6.9101 |
| | 0.8 | 5.0048 | 5.0048 | 6.85E-07 | 6.85E-07 | 3.9153 | 3.9153 | 5.5586 | 5.5586 |
| | 0.8 | 40.0384 | 40.0384 | 5.48E-06 | 5.48E-06 | 3.9153 | 3.9153 | 5.5586 | 5.5586 |
| | 1 | 51.2992 | 38.7872 | 3.60E-06 | 3.60E-06 | 3.9409 | 3.89 | 5.2009 | 5.2115 |
| | 0.8 | 12.512 | 12.512 | 1.39E-06 | 1.39E-06 | 3.9485 | 3.9485 | 5.3675 | 5.3675 |
| | 0.4 | 185.1776 | 187.68 | 1.62E-05 | 2.21E-05 | 0.335 | 0.4291 | 7.2083 | 7.2307 |
| | 1 | 32.5312 | 32.5312 | 2.08E-06 | 2.08E-06 | 3.9378 | 3.9378 | 5.2313 | 5.2313 |
| | 1 | 815.7824 | 815.7824 | 8.83E-05 | 8.83E-05 | 1.469 | 1.469 | 5.3261 | 5.3261 |
| | 1 | 73.8208 | 40.0384 | 1.35E-05 | 5.48E-06 | 0.1141 | 3.9153 | 9.3203 | 5.5586 |
| | 0.4 | 43.792 | 58.8064 | 2.67E-06 | 3.70E-06 | 3.9446 | 3.9487 | 5.2209 | 5.2227 |
| | 0.8 | 240.2304 | 30.0288 | 2.07E-06 | 2.68E-07 | 3.9395 | 3.8045 | 4.6738 | 3.8393 |
| | 0.4 | 78.8256 | 90.0864 | 1.35E-05 | 1.38E-05 | 0.1315 | 0.1315 | 7.2451 | 7.2451 |
| | 0.4 | 3209.328 | 1555.2416 | 2.71E-04 | 1.47E-04 | 0.2087 | 2.29E-01 | 6.2969 | 7.0092 |
| | 0.8 | 28.7776 | 28.7776 | 1.32E-06 | 1.32E-06 | 4.0661 | 4.0661 | 5.1155 | 5.1155 |
| | 1 | 36.2848 | 73.8208 | 2.71E-06 | 1.35E-05 | 3.9378 | 0.1141 | 5.2313 | 6.9997 |
| | 1 | 15.0144 | 22.5216 | 1.11E-06 | 1.35E-05 | 4.051 | 4.0503 | 5.2854 | 5.1241 |
| | 1 | 96.3424 | 113.8592 | 1.44E-05 | 1.59E-05 | 0.2616 | 0.3507 | 7.2457 | 7.2395 |
| | 1 | 1051.008 | 1051.008 | 3.78E-05 | 3.78E-05 | 4.1483 | 4.1483 | 3.2117 | 3.2117 |
| | 0.8 | 40.0384 | 40.0384 | 3.88E-05 | 3.88E-05 | 3.9153 | 3.9153 | 5.5586 | 5.5586 |
| Multimodal | 0.4 | 5.0048 | 5.0048 | 6.85E-07 | 6.85E-07 | 3.9153 | 3.9153 | 5.5586 | 5.5586 |
| | 1 | 20.0192 | 20.0192 | 2.74E-06 | 2.74E-06 | 3.9153 | 3.9153 | 5.5492 | 5.5492 |
| | 0.4 | 1886.8096 | 1886.8096 | 1.41E-04 | 1.41E-04 | 0.2326 | 0.2326 | 6.7635 | 6.7635 |
| | 0.6 | 6.256 | 6.256 | 6.35E-07 | 6.35E-07 | 3.8426 | 3.8426 | 5.4372 | 5.4372 |
| | 0.6 | 6.256 | 8.7584 | 6.93E-07 | 7.38E-07 | 3.9485 | 3.9895 | 5.3708 | 5.452 |
| | 1 | 36.2848 | 36.2848 | 7.16E-06 | 7.16E-06 | 0.3785 | 0.3785 | 7.2871 | 7.2871 |
| | 1 | 26.2752 | 26.2752 | 4.85E-06 | 4.85E-06 | 1.4197 | 1.4197 | 7.2451 | 7.2451 |
| | 1 | 60.0576 | 85.0816 | 9.36E-06 | 2.02E-05 | 0.1315 | 0.2648 | 7.2451 | 7.2284 |
| | 1 | 120.1152 | 120.1152 | 5.19E-06 | 5.19E-06 | 3.9058 | 3.9058 | 4.7301 | 4.7301 |
| | 1 | 63.8112 | 121.3664 | 9.47E-06 | 1.59E-05 | 0.2152 | 0.2224 | 7.0874 | 7.2451 |

Table 9: PPA Info of GPT4 on Part of Generated Design

| Modal | Function Correctness | Area | | Power | | Hold WNS | | Setup TNS | |
|---|---|---|---|---|---|---|---|---|---|
| | | Generated | Reference | Generated | Reference | Generated | Reference | Generated | Reference |
| Text | 0.6 | 6.256 | 3.7536 | 6.33E-07 | 5.92E-07 | 3.8839 | 3.8395 | 5.5943 | 5.6193 |
| | 1 | 7.5072 | 5.0048 | 7.01E-07 | 6.85E-07 | 3.9746 | 3.9153 | 5.504 | 5.5586 |
| | 0.2 | 6.256 | 6.256 | 6.93E-07 | 6.93E-07 | 3.9485 | 3.9485 | 5.3708 | 5.3708 |
| | 0.8 | 22.5216 | 22.5216 | 1.63E-06 | 1.63E-06 | 3.8877 | 3.8877 | 5.317 | 5.317 |
| | 0.2 | 22.5216 | 22.5216 | 1.63E-06 | 1.63E-06 | 3.8877 | 3.8877 | 5.317 | 5.317 |
| | 0.8 | 5.0048 | 5.0048 | 6.85E-07 | 6.85E-07 | 3.9153 | 3.9153 | 5.5586 | 5.5586 |
| | 0.6 | 40.0384 | 40.0384 | 5.48E-06 | 5.48E-06 | 3.9153 | 3.9153 | 5.5586 | 5.5586 |
| | 1 | 51.2992 | 38.7872 | 3.60E-06 | 3.60E-06 | 3.9409 | 3.89 | 5.2009 | 5.2115 |
| | 0.8 | 12.512 | 12.512 | 1.39E-06 | 1.39E-06 | 3.9485 | 3.9485 | 5.3675 | 5.3675 |
| | 0.4 | 171.4144 | 187.68 | 1.62E-05 | 2.21E-05 | 0.4056 | 0.4291 | 7.2206 | 7.2307 |
| | 1 | 32.5312 | 32.5312 | 2.08E-06 | 2.08E-06 | 3.9378 | 3.9378 | 5.2313 | 5.2313 |
| | 1 | 815.7824 | 815.7824 | 8.83E-05 | 8.83E-05 | 1.469 | 1.469 | 5.3261 | 5.3261 |
| | 1 | 40.0384 | 40.0384 | 5.48E-06 | 5.48E-06 | 3.9153 | 3.9153 | 5.5586 | 5.5586 |
| | 0.4 | 53.8016 | 58.8064 | 3.68E-06 | 3.70E-06 | 3.9412 | 3.9487 | 5.2008 | 5.2227 |
| | 0.8 | 30.0288 | 30.0288 | 2.68E-07 | 2.68E-07 | 3.8045 | 3.8045 | 3.8393 | 3.8393 |
| | 0.4 | 21550.6688 | 22096.192 | 3.79E-03 | 4.61E-03 | 0.2104 | 0.2104 | 3.8231 | 3.7868 |
| | 0.8 | 1068.5248 | 1555.2416 | 1.34E-04 | 1.47E-04 | 0.2395 | 0.229 | 6.9484 | 7.0092 |
| | 0.6 | 17.5168 | 22.5216 | 1.32E-06 | 1.32E-06 | 3.8788 | 4.0503 | 5.3341 | 5.1241 |
| | 1 | 122.6176 | 122.6176 | 1.30E-05 | 1.30E-05 | 1.4344 | 1.4344 | 7.2451 | 7.2451 |
| | 1 | 96.3424 | 113.8592 | 1.44E-05 | 1.59E-05 | 0.2616 | 0.3507 | 7.2451 | 7.2395 |
| | 0.8 | 11.2608 | 11.2608 | 1.03E-06 | 1.03E-06 | 4.051 | 4.051 | 5.2878 | 5.2878 |
| | 1 | 1051.008 | 1051.008 | 3.78E-05 | 3.78E-05 | 4.1483 | 4.1483 | 3.2117 | 3.2117 |
| | 0.8 | 210.2016 | 40.0384 | 3.88E-05 | 3.88E-05 | 1.469 | 3.9153 | 7.2451 | 5.5586 |
| Multimodal | 1 | 5.0048 | 5.0048 | 6.85E-07 | 6.85E-07 | 3.9153 | 3.9153 | 5.5586 | 5.5586 |
| | 1 | 20.0192 | 20.0192 | 2.74E-06 | 2.74E-06 | 3.9153 | 3.9153 | 5.5492 | 5.5492 |
| | 1 | 36.2848 | 36.2848 | 7.16E-06 | 7.16E-06 | 0.3785 | 0.3785 | 7.2871 | 7.2871 |
| | 1 | 26.2752 | 26.2752 | 4.85E-06 | 4.85E-06 | 1.4197 | 1.4197 | 7.2451 | 7.2451 |
| | 1 | 60.0576 | 85.0816 | 9.36E-06 | 2.02E-05 | 0.1315 | 0.2648 | 7.2451 | 7.2284 |
| | 1 | 91.3376 | 120.1152 | 5.29E-06 | 5.19E-06 | 3.8815 | 3.9058 | 4.5263 | 4.7301 |
| | 0.6 | 85.0816 | 121.3664 | 1.38E-05 | 1.59E-05 | 0.2737 | 0.2224 | 7.0185 | 7.2451 |

Table 10: Results of Tested Models.

| | model | Knowledge Mastery | Knowledge Transfer | Debugging | Function | Synatx | Synthesizbility |
|---|---|---|---|---|---|---|---|
| GenBen-all | gpt-4-turbo | 57.00% | 62.00% | 40.00% | 21.20% | 100.00% | 93.70% |
| GenBen-allmodal-L1 | gpt-4-turbo | 64.00% | 70.00% | 37.70% | 30.90% | 100.00% | 90.70% |
| GenBen-allmodal-L2 | gpt-4-turbo | 56.00% | 65.00% | 33.30% | 24.20% | 99.40% | 96.40% |
| GenBen-allmodal-L3 | gpt-4-turbo | 52.00% | 52.00% | 21.10% | 9.10% | 98.90% | 92.40% |
| GenBen-mm | gpt-4-turbo | 27.00% | 67.00% | 63.30% | 16.70% | 100.00% | 96.50% |
| GenBen-mm-L1 | gpt-4-turbo | 0.00% | 67.00% | 55.00% | 24.30% | 100.00% | 95.00% |
| GenBen-mm-L2 | gpt-4-turbo | 40.00% | 100.00% | 50.00% | 20.10% | 99.40% | 97.50% |
| GenBen-mm-L3 | gpt-4-turbo | 40.00% | 33.00% | 10.00% | 8.20% | 99.40% | 97.50% |
| GenBen-text | gpt-4-turbo | 65.00% | 62.00% | 35.60% | 21.30% | 100.00% | 89.80% |
| GenBen-text-L1 | gpt-4-turbo | 80.00% | 70.00% | 33.30% | 20.90% | 100.00% | 83.20% |
| GenBen-text-L2 | gpt-4-turbo | 60.00% | 60.00% | 30.00% | 23.90% | 100.00% | 95.60% |
| GenBen-text-L3 | gpt-4-turbo | 55.00% | 55.00% | 26.60% | 16.50% | 100.00% | 82.80% |
| GenBen-all | gpt-4o | 69.00% | 71.00% | 52.20% | 34.80% | 100.00% | 96.90% |
| GenBen-allmodal-L1 | gpt-4o | 72.00% | 83.00% | 43.20% | 38.60% | 100.00% | 94.60% |
| GenBen-allmodal-L2 | gpt-4o | 64.00% | 74.00% | 38.40% | 32.60% | 100.00% | 98.80% |
| GenBen-allmodal-L3 | gpt-4o | 72.00% | 57.00% | 34.20% | 29.50% | 100.00% | 99.40% |
| GenBen-mm | gpt-4o | 47.00% | 78.00% | 71.70% | 37.50% | 100.00% | 100.00% |
| GenBen-mm-L1 | gpt-4o | 40.00% | 67.00% | 80.00% | 37.50% | 100.00% | 95.00% |
| GenBen-mm-L2 | gpt-4o | 60.00% | 100.00% | 70.00% | 32.50% | 100.00% | 97.50% |
| GenBen-mm-L3 | gpt-4o | 40.00% | 67.00% | 35.00% | 28.50% | 100.00% | 97.50% |
| GenBen-text | gpt-4o | 75.00% | 70.00% | 40.00% | 32.00% | 97.50% | 96.00% |
| GenBen-text-L1 | gpt-4o | 80.00% | 85.00% | 33.30% | 34.70% | 95.00% | 95.00% |
| GenBen-text-L2 | gpt-4o | 65.00% | 70.00% | 30.00% | 30.50% | 97.50% | 97.50% |
| GenBen-text-L3 | gpt-4o | 80.00% | 55.00% | 36.70% | 27.50% | 100.00% | 97.50% |
| GenBen-text | gpt-3.5-turbo | 63.00% | 60.00% | 37.80% | 26.70% | 98.10% | 93.30% |
| GenBen-text-L1 | gpt-3.5-turbo | 65.00% | 50.00% | 46.70% | 29.00% | 92.00% | 72.00% |
| GenBen-text-L2 | gpt-3.5-turbo | 65.00% | 60.00% | 26.70% | 24.00% | 100.00% | 96.80% |
| GenBen-text-L3 | gpt-3.5-turbo | 60.00% | 70.00% | 24.70% | 19.00% | 99.20% | 87.20% |
| GenBen-all | claude3.5 | 59.00% | 61.00% | 55.40% | 35.40% | 98.60% | 90.00% |
| GenBen-allmodal-L1 | claude3.5 | 64.00% | 70.00% | 53.70% | 43.30% | 97.00% | 86.70% |
| GenBen-allmodal-L2 | claude3.5 | 56.00% | 65.00% | 48.90% | 37.10% | 100.00% | 100.00% |
| GenBen-allmodal-L3 | claude3.5 | 56.00% | 48.00% | 33.70% | 28.50% | 98.80% | 87.30% |
| GenBen-mm | claude3.5 | 47.00% | 44.00% | 55.00% | 39.20% | 100.00% | 92.50% |
| GenBen-mm-L1 | claude3.5 | 20.00% | 67.00% | 55.00% | 45.00% | 100.00% | 90.00% |
| GenBen-mm-L2 | claude3.5 | 60.00% | 67.00% | 45.00% | 35.00% | 100.00% | 100.00% |
| GenBen-mm-L3 | claude3.5 | 60.00% | 0.00% | 35.00% | 37.50% | 100.00% | 87.50% |
| GenBen-text | claude3.5 | 62.00% | 63.00% | 55.60% | 22.10% | 98.10% | 89.10% |
| GenBen-text-L1 | claude3.5 | 75.00% | 70.00% | 53.30% | 21.60% | 96.00% | 80.80% |
| GenBen-text-L2 | claude3.5 | 55.00% | 65.00% | 50.00% | 19.20% | 100.00% | 99.20% |
| GenBen-text-L3 | claude3.5 | 55.00% | 55.00% | 33.30% | 25.60% | 98.40% | 87.20% |
| GenBen-text | llama3 | 68.00% | 70.00% | 40.00% | 6.90% | 85.90% | 57.30% |
| GenBen-text-L1 | llama3 | 75.00% | 75.00% | 53.30% | 6.10% | 78.40% | 56.00% |
| GenBen-text-L2 | llama3 | 70.00% | 70.00% | 43.30% | 6.40% | 89.60% | 58.40% |
| GenBen-text-L3 | llama3 | 60.00% | 65.00% | 6.67% | 7.20% | 89.60% | 57.40% |
| GenBen-all | qwen-vl-max | 59.00% | 55.00% | 36.50% | 26.50% | 88.60% | 78.90% |
| GenBen-allmodal-L1 | qwen-vl-max | 72.00% | 74.00% | 43.20% | 29.90% | 84.20% | 78.20% |
| GenBen-allmodal-L2 | qwen-vl-max | 52.00% | 57.00% | 40.70% | 26.50% | 95.20% | 87.30% |
| GenBen-allmodal-L3 | qwen-vl-max | 52.00% | 35.00% | 23.20% | 22.20% | 86.20% | 71.30% |
| GenBen-mm | qwen-vl-max | 53.00% | 89.00% | 55.00% | 49.30% | 100.00% | 91.70% |
| GenBen-mm-L1 | qwen-vl-max | 60.00% | 100.00% | 55.00% | 62.50% | 100.00% | 100.00% |
| GenBen-mm-L2 | qwen-vl-max | 40.00% | 100.00% | 45.00% | 51.20% | 100.00% | 100.00% |
| GenBen-mm-L3 | qwen-vl-max | 60.00% | 67.00% | 35.00% | 25.00% | 100.00% | 87.50% |
| GenBen-text | qwen-vl-max | 60.00% | 50.00% | 44.40% | 20.20% | 84.80% | 76.90% |
| GenBen-text-L1 | qwen-vl-max | 75.00% | 70.00% | 40.00% | 22.80% | 79.20% | 75.20% |
| GenBen-text-L2 | qwen-vl-max | 55.00% | 50.00% | 43.00% | 22.40% | 93.60% | 86.40% |
| GenBen-text-L3 | qwen-vl-max | 50.00% | 30.00% | 20.00% | 21.30% | 81.90% | 69.30% |
| GenBen-all | qwen-vl-plus | 45.00% | 46.00% | 32.60% | 16.30% | 78.40% | 66.40% |
| GenBen-allmodal-L1 | qwen-vl-plus | 52.00% | 52.00% | 32.60% | 20.00% | 78.80% | 65.50% |
| GenBen-allmodal-L2 | qwen-vl-plus | 40.00% | 43.00% | 27.90% | 16.00% | 85.50% | 74.50% |
| GenBen-allmodal-L3 | qwen-vl-plus | 44.00% | 43.00% | 7.40% | 12.00% | 71.30% | 59.30% |
| GenBen-mm | qwen-vl-plus | 20.00% | 44.00% | 8.30% | 4.20% | 58.30% | 33.30% |
| GenBen-mm-L1 | qwen-vl-plus | 0.00% | 67.00% | 5.00% | 0.00% | 77.50% | 35.00% |
| GenBen-mm-L2 | qwen-vl-plus | 40.00% | 33.00% | 0.00% | 12.50% | 60.00% | 37.50% |
| GenBen-mm-L3 | qwen-vl-plus | 20.00% | 33.00% | 0.00% | 0.00% | 37.50% | 27.50% |
| GenBen-text | qwen-vl-plus | 52.00% | 47.00% | 44.40% | 20.20% | 84.90% | 76.90% |
| GenBen-text-L1 | qwen-vl-plus | 65.00% | 50.00% | 40.00% | 22.80% | 79.20% | 75.20% |
| GenBen-text-L2 | qwen-vl-plus | 40.00% | 45.00% | 43.30% | 17.40% | 93.60% | 86.40% |
| GenBen-text-L3 | qwen-vl-plus | 50.00% | 45.00% | 20.00% | 16.30% | 81.90% | 69.30% |
| GenBen-all | GLM-4V-plus | 51.00% | 62.00% | 39.60% | 12.50% | 71.70% | 51.10% |
| GenBen-allmodal-L1 | GLM-4V-plus | 60.00% | 65.00% | 43.20% | 15.50% | 67.30% | 40.00% |
| GenBen-allmodal-L2 | GLM-4V-plus | 44.00% | 74.00% | 17.40% | 13.20% | 65.10% | 42.80% |
| GenBen-allmodal-L3 | GLM-4V-plus | 48.00% | 48.00% | 28.40% | 8.00% | 83.10% | 70.40% |
| GenBen-mm | GLM-4V-plus | 27.00% | 89.00% | 30.00% | 28.30% | 90.80% | 69.20% |
| GenBen-mm-L1 | GLM-4V-plus | 20.00% | 100.00% | 30.00% | 17.50% | 77.50% | 47.50% |
| GenBen-mm-L2 | GLM-4V-plus | 20.00% | 100.00% | 25.00% | 36.50% | 95.10% | 61.00% |
| GenBen-mm-L3 | GLM-4V-plus | 40.00% | 67.00% | 35.00% | 30.00% | 100.00% | 97.50% |
| GenBen-text | GLM-4V-plus | 57.00% | 58.00% | 42.20% | 7.50% | 65.60% | 45.30% |
| GenBen-text-L1 | GLM-4V-plus | 70.00% | 60.00% | 46.70% | 11.00% | 64.00% | 37.60% |
| GenBen-text-L2 | GLM-4V-plus | 50.00% | 70.00% | 23.30% | 5.60% | 55.20% | 36.80% |
| GenBen-text-L3 | GLM-4V-plus | 50.00% | 45.00% | 26.70% | 5.00% | 77.80% | 61.90% |
| GenBen-text | GLM-4 | 57.00% | 58.00% | 42.20% | 17.50% | 65.60% | 45.30% |
| GenBen-text-L1 | GLM-4 | 50.00% | 25.00% | 33.30% | 24.80% | 84.00% | 76.00% |
| GenBen-text-L2 | GLM-4 | 45.00% | 45.00% | 43.30% | 19.00% | 95.20% | 94.40% |
| GenBen-text-L3 | GLM-4 | 50.00% | 20.00% | 6.70% | 13.00% | 96.00% | 72.80% |

The result is shown in Table 10. This provides a statistical analysis of the tested models, covering knowledge master, knowledge transfer, debugging, functional correctness, syntax correctness, and synthesizability. For further QoR analysis, data from the best-performing models, GPT-4o and Claude 3.5, are included in the main text.

The data in the table demonstrate the effectiveness of task categorization, the necessity of synthesizability metrics, and the correlation between knowledge points and coding abilities, aligning with the benchmark's design expectations.

### A.7 TUTORIAL: EVALUATING LLM PERFORMANCE WITH GENBEN

You can access the complete GenBen code via the following link: GenBen Repository. This guide will walk you through evaluating the performance of Large Language Models (LLMs) in hardware design and obtaining detailed results using the command line.

#### A.7.1 STEP-BY-STEP INSTRUCTIONS

**Clone the GenBen Repository**

First, clone the GenBen repository to your local machine:

```
git clone https://anonymous.4open.science/r/GENBEN-2812
cd GENBEN-2812
```

**Run the Evaluation Script**

Using the command line, you can evaluate the performance of LLMs with the following command:

```
python genben.py --mode all --model gpt4
```

This command runs the evaluation with the specified parameters.

**Understanding the Command Parameters**

- `--mode`: This parameter controls the type of tasks input into the LLMs. There are three available options:
  - `all`: Enables the input of all task types.
  - `mm`: Allows for multi-modal tasks.
  - `text`: Restricts the input to text-based tasks only.
- `--model`: This parameter specifies the model of the LLMs. Adjust this parameter according to the specific API of the LLMs you are using.

Example:

```
python genben.py --mode text --model gpt4
```

This command evaluates the gpt4 model using only text-based tasks.

#### A.7.2 REFER TO THE README FOR DETAILED INSTRUCTIONS

For more detailed usage instructions, please refer to the README file included in the GenBen project. The README file contains comprehensive information

### A.8 OPEN SOURCE DECLARATION

To foster transparency, collaboration, and innovation, the GenBen benchmark will be released under the **MIT** open-source license. This ensures that researchers, educators, and practitioners can freely access, use, modify, and distribute the benchmark without any restrictions.

Upon the completion of the peer-review process, the full dataset, along with all associated scripts and documentation, will be made publicly available. We hope to support the global research community in advancing the field of hardware design and AI-driven EDA.

