# OpenReview forum: "GenBen:A Genarative Benchmark for LLM-Aided Design"
_ICLR.cc/2025/Conference — Submitted to ICLR 2025_

### Official Review · Reviewer_Ta3C · 2024-11-02

**Soundness:** 3
**Presentation:** 2
**Contribution:** 2
**Rating:** 6
**Confidence:** 3

**Summary:**

The paper introduces GenBen, a generative benchmark designed to evaluate the performance of large language models in hardware design automation. As LLM-aided design progresses rapidly, it has become essential to assess these models' efficacy in automating various stages of hardware design, from high-level architecture to low-level circuit optimization. GenBen distinguishes itself from existing benchmarks by emphasizing aspects such as Quality of Result metrics, design diversity, multimodality, and data contamination prevention.

**Strengths:**

1.  GenBen implements an end-to-end verification process, ensuring functional coverage for RTL designs, which enhances the reliability of generated designs.

2. With a dataset derived from silicon-proven projects, textbooks, and community sources, GenBen categorizes tasks into three difficulty levels, allowing for more granular assessments.

3. By employing static and dynamic perturbations, GenBen mitigates the risk of data leakage from pre-training datasets, maintaining the integrity of evaluation results.

**Weaknesses:**

1. I tried to access the anonymous link provided by the authors for the benchmark, but it seems to be unavailable. Could the authors provide a working link during the rebuttal phase? This would allow me to evaluate the quality of the dataset

2. Another minor concern: Although I really appreciate the multimodal and contamination-free dataset, I personally feel that this paper might be better suited for a hardware-focused conference.

**Questions:**

See Weakness.

---

### Official Review · Reviewer_iqgx · 2024-11-02

**Soundness:** 2
**Presentation:** 3
**Contribution:** 2
**Rating:** 5
**Confidence:** 5

**Summary:**

The paper introduces GenBen, a benchmark for LLM-aided design (LAD) specifically targeted at evaluating the performance of large language models (LLMs) in hardware design. This benchmark covers a diverse set of tasks with both unimodal and multimodal support, drawing from 20 sources across four primary domains, further organized into five detailed categories. The tasks are structured across three difficulty tiers, aiming to comprehensively assess models' skills in LAD. The paper also addresses issues of data contamination, dataset diversity, and incorporates a pass@5 evaluation setup across nine models, assessed with comprehensive metrics for syntactic, functional, and Quality-of-Results (QoR) evaluation. The benchmark and its workflow are structured to support reproducibility, utilizing open-source tools like Icarus Verilog and OpenLane.

**Strengths:**

1.	Comprehensive Metric Coverage: GenBen includes both syntactic and functional correctness metrics, as well as QoR metrics (synthesizability, power, area, timing), giving a well-rounded evaluation of LLMs in LAD. The use of open-source tools like Icarus Verilog and OpenLane (as detailed in Section 3.6 and Appendix A.3.1) enhances reproducibility.
2.	Perturbation Strategy: GenBen employs both surface and semantic perturbations to prevent test set contamination, a thoughtful design that diversifies test inputs dynamically, as discussed in Section 3.4 and illustrated in Figure 2. This strategy enhances benchmark reliability by mitigating memorization effects.
3.	Multimodal Support: Recognizing the multimodal nature of hardware design, GenBen supports tasks that include textual, schematic, and architectural data (Section 3.5). This addition is crucial for practical LAD assessments where real-world design tasks often require multimodal inputs.
4.	Clear Evaluation Pipeline: The step-by-step pipeline in Figure 2 provides clear guidance for users, with explicit instructions on test construction, perturbation, and evaluation metrics (Section 3.2). The modular approach allows GenBen to assess models on multiple metrics, making it adaptable to diverse LAD scenarios.
5.	Thorough Documentation: The appendices provide exhaustive details on LAD-related concepts, evaluation metrics, and the role of open-source tools (Appendix A.1 to A.3), which can aid users in implementing the benchmark effectively.

**Weaknesses:**

The paper is not much relevant to the ICLR community, and does not provide a strong mathematical foundation. A more suitable venue for this paper could be ICLAD Conference or some Design Automation Conference like DAC or ICCAD.

1.	Data Imbalance in Test Categories: Table 3 presents disparities across task categories (e.g., 99 Design tasks vs. 57 Debug tasks). While task diversity is commendable, the authors should clarify why such an imbalance exists and discuss potential implications for model performance assessment, especially if specific categories are overrepresented. An explanation of how this might affect the reliability of the benchmark would strengthen the dataset's construction rationale (Section 3.3.3).
2.	Model Selection Motivation: Section 4.1 introduces nine models across five families but lacks justification for these particular choices. Clarifying why specific models (e.g., GPT, Claude, LLaMA, QWEN, and GLM families) were included could strengthen the argument, especially if these models are particularly relevant for LAD tasks. For instance, what LAD-specific capabilities do these models bring? Including performance characteristics in multimodal tasks or LAD-related features in each model would better contextualize the selection.
3.	Difficulty Tiering: The paper divides tasks into three difficulty levels (L1 to L3) with proper descriptions (Table 2). However, detailing how each tier’s tasks correspond to LAD challenges (e.g., debugging complexity or resource optimization) would add valuable context. The authors could enhance Appendix A.4 by providing specific examples that illustrate the distinctions between L1, L2, and L3, particularly within each task type (knowledge, design, debugging). Quantitative data on average model performance per tier might also reveal trends that could guide future benchmarking improvements.
4.	Detailed Case Study on Perturbation: The perturbation strategy is briefly explained in Section 3.4, with an overview of surface and semantic techniques. However, a specific example, especially for semantic perturbation, would make this section clearer. Adding a detailed case study would provide a more straightforward demonstration of how the data were perturbed, also for semantic perturbation, due to the randomness, is there a manageable gap of difficulty to change? For instance, showing a task before and after both types of perturbation would help readers understand the differences in difficulty introduced by each type. Furthermore, since semantic perturbation could change task difficulty, addressing how fairness is maintained would strengthen the case for using this approach.
5.	Elaboration on Workflow: The "Dynamic Test Kit" component in Figure 2 and Section 3.2 would benefit from additional detail. For instance, does the Dynamic Test Kit contain tools that adjust task difficulty, or is it limited to generating variants of test inputs? Furthermore, clarifying the connection between the scoring system described in Section 4.1 and the metrics reported in Table 5 could make the scoring more interpretable. Specifically, explaining if the pass rate is calculated per task or per difficulty level would aid in understanding the benchmark's scoring logic.
6.	Experiment Results Interpretation: Section 4.2 and Figures 4-12 illustrate experiment results, but a breakdown of model performance by metric (e.g., Syntax, Synthesizability, Function) would be helpful. Discussing why Syntax scores are generally higher while Function scores lag behind could provide insight into specific model weaknesses. Additionally, including a statistical analysis (e.g., variance in pass rates across difficulty levels) would offer a clearer picture of each model’s strengths and limitations.
7.	Supporting Literature for Multimodal Justification: Section 3.5 could be enhanced by citing specific LAD scenarios or prior research that demonstrate the need for multimodal tasks (e.g., combining schematics with HDL code) to support the claim “This feature is particularly important because real-world design processes often require the integration of various forms of data, such as textual specifications, diagrams, and architectural schematics”. Including practical examples from hardware design projects that require multimodal support would illustrate the real-world relevance of GenBen’s multimodal component. This would also substantiate the statement that real-world LAD tasks are inherently multimodal.
9.	Future Work and Limitations: The paper would benefit from a Limitations and Future Work section, discussing GenBen's adaptability to emerging LAD requirements (e.g., integration with more complex EDA tools or AI-driven hardware design) could provide a roadmap for GenBen’s evolution. Additionally, mentioning scalability limitations (e.g., handling larger designs or more complex verification tasks) would provide a balanced view of the benchmark’s potential applications and restrictions.
10.	Evaluation Metric Granularity: While Table 5 provides an overview of evaluation metrics, further clarification on the weighting or relative importance of these metrics (e.g., Syntax vs. Synthesizability vs. QoR metrics) could improve transparency. Explaining if QoR metrics are prioritized over syntactic correctness for certain tasks, such as those targeting manufacturable designs, would give a better understanding of how the benchmark defines "success." Additionally, addressing any trade-offs between these metrics, especially in cases where models might excel in one metric while underperforming in others, would be helpful (e.g., does a high Syntax score compensate for lower Synthesizability?).
11.	Benchmark Customizability: GenBen is described as an end-to-end LAD benchmark. A brief mention of how the benchmark might be tailored for different LAD applications (e.g., specialized RTL verification or energy optimization) in Section 3.2 would clarify the benchmark’s flexibility.
12.	Clearer Description of Multimodal Task Processing: Although Section 3.5 explains the importance of multimodal tasks, more detail on how GenBen processes or evaluates multimodal inputs would be beneficial. For instance, specifying how different data forms (e.g., schematics vs. HDL code) are presented to models and if specific evaluation criteria are adapted to multimodal tasks could provide clarity. Addressing whether the models are expected to interpret these inputs sequentially or concurrently, and if multimodal inputs impact the difficulty tiering, would improve understanding of the benchmark's multimodal handling.
13.	Clarification of Pass@5 Evaluation and Scoring: The pass@5 scoring strategy in Section 4.1 is introduced without much context on its relevance to LAD tasks. Explaining why pass@5 is chosen over other scoring metrics (e.g., pass@10, exact match) and how it aligns with real-world LAD evaluation (e.g., tolerance for minor errors in preliminary passes) would strengthen its justification. Additionally, describing if scoring varies by task difficulty or complexity would make the scoring methodology clearer.

**Questions:**

Address the questions raised in the weaknesses, as mentioned below.

1.	Data Imbalance in Test Categories: Table 3 presents disparities across task categories (e.g., 99 Design tasks vs. 57 Debug tasks). While task diversity is commendable, the authors should clarify why such an imbalance exists and discuss potential implications for model performance assessment, especially if specific categories are overrepresented. An explanation of how this might affect the reliability of the benchmark would strengthen the dataset's construction rationale (Section 3.3.3).
2.	Model Selection Motivation: Section 4.1 introduces nine models across five families but lacks justification for these particular choices. Clarifying why specific models (e.g., GPT, Claude, LLaMA, QWEN, and GLM families) were included could strengthen the argument, especially if these models are particularly relevant for LAD tasks. For instance, what LAD-specific capabilities do these models bring? Including performance characteristics in multimodal tasks or LAD-related features in each model would better contextualize the selection.
3.	Difficulty Tiering: The paper divides tasks into three difficulty levels (L1 to L3) with proper descriptions (Table 2). However, detailing how each tier’s tasks correspond to LAD challenges (e.g., debugging complexity or resource optimization) would add valuable context. The authors could enhance Appendix A.4 by providing specific examples that illustrate the distinctions between L1, L2, and L3, particularly within each task type (knowledge, design, debugging). Quantitative data on average model performance per tier might also reveal trends that could guide future benchmarking improvements.
4.	Detailed Case Study on Perturbation: The perturbation strategy is briefly explained in Section 3.4, with an overview of surface and semantic techniques. However, a specific example, especially for semantic perturbation, would make this section clearer. Adding a detailed case study would provide a more straightforward demonstration of how the data were perturbed, also for semantic perturbation, due to the randomness, is there a manageable gap of difficulty to change? For instance, showing a task before and after both types of perturbation would help readers understand the differences in difficulty introduced by each type. Furthermore, since semantic perturbation could change task difficulty, addressing how fairness is maintained would strengthen the case for using this approach.
5.	Elaboration on Workflow: The "Dynamic Test Kit" component in Figure 2 and Section 3.2 would benefit from additional detail. For instance, does the Dynamic Test Kit contain tools that adjust task difficulty, or is it limited to generating variants of test inputs? Furthermore, clarifying the connection between the scoring system described in Section 4.1 and the metrics reported in Table 5 could make the scoring more interpretable. Specifically, explaining if the pass rate is calculated per task or per difficulty level would aid in understanding the benchmark's scoring logic.
6.	Experiment Results Interpretation: Section 4.2 and Figures 4-12 illustrate experiment results, but a breakdown of model performance by metric (e.g., Syntax, Synthesizability, Function) would be helpful. Discussing why Syntax scores are generally higher while Function scores lag behind could provide insight into specific model weaknesses. Additionally, including a statistical analysis (e.g., variance in pass rates across difficulty levels) would offer a clearer picture of each model’s strengths and limitations.
7.	Supporting Literature for Multimodal Justification: Section 3.5 could be enhanced by citing specific LAD scenarios or prior research that demonstrate the need for multimodal tasks (e.g., combining schematics with HDL code) to support the claim “This feature is particularly important because real-world design processes often require the integration of various forms of data, such as textual specifications, diagrams, and architectural schematics”. Including practical examples from hardware design projects that require multimodal support would illustrate the real-world relevance of GenBen’s multimodal component. This would also substantiate the statement that real-world LAD tasks are inherently multimodal.
9.	Future Work and Limitations: The paper would benefit from a Limitations and Future Work section, discussing GenBen's adaptability to emerging LAD requirements (e.g., integration with more complex EDA tools or AI-driven hardware design) could provide a roadmap for GenBen’s evolution. Additionally, mentioning scalability limitations (e.g., handling larger designs or more complex verification tasks) would provide a balanced view of the benchmark’s potential applications and restrictions.
10.	Evaluation Metric Granularity: While Table 5 provides an overview of evaluation metrics, further clarification on the weighting or relative importance of these metrics (e.g., Syntax vs. Synthesizability vs. QoR metrics) could improve transparency. Explaining if QoR metrics are prioritized over syntactic correctness for certain tasks, such as those targeting manufacturable designs, would give a better understanding of how the benchmark defines "success." Additionally, addressing any trade-offs between these metrics, especially in cases where models might excel in one metric while underperforming in others, would be helpful (e.g., does a high Syntax score compensate for lower Synthesizability?).
11.	Benchmark Customizability: GenBen is described as an end-to-end LAD benchmark. A brief mention of how the benchmark might be tailored for different LAD applications (e.g., specialized RTL verification or energy optimization) in Section 3.2 would clarify the benchmark’s flexibility.
12.	Clearer Description of Multimodal Task Processing: Although Section 3.5 explains the importance of multimodal tasks, more detail on how GenBen processes or evaluates multimodal inputs would be beneficial. For instance, specifying how different data forms (e.g., schematics vs. HDL code) are presented to models and if specific evaluation criteria are adapted to multimodal tasks could provide clarity. Addressing whether the models are expected to interpret these inputs sequentially or concurrently, and if multimodal inputs impact the difficulty tiering, would improve understanding of the benchmark's multimodal handling.
13.	Clarification of Pass@5 Evaluation and Scoring: The pass@5 scoring strategy in Section 4.1 is introduced without much context on its relevance to LAD tasks. Explaining why pass@5 is chosen over other scoring metrics (e.g., pass@10, exact match) and how it aligns with real-world LAD evaluation (e.g., tolerance for minor errors in preliminary passes) would strengthen its justification. Additionally, describing if scoring varies by task difficulty or complexity would make the scoring methodology clearer.

Presentation Comments:

1.	Typos: The misspellings in Figures 4-12 ("synatx" and "synthesizbility") should be corrected to "syntax" and "synthesizability" respectively.
2.	Formula clarification in Section 4.1: The term "a" should be clearly described in Formula (1).
3.	Font Size: Increase the font size in Figures 13, 14, and 16 to improve the readability of numerical data.
4.	Unnecessary space: There is some empty space after Table 3 due to Figure 3 which should be fixed.
5.	Acronym definitions: Define acronyms like LAD, QoR, HDL, and EDA at their first appearance in the main text (even if they are defined in the abstract or appendix).
6.	Cross-Referencing Appendix Sections: In the main text, add references to specific appendix sections where supplementary information is provided (e.g., "See Appendix A.4 for detailed examples of difficulty tiers"). This will guide readers directly to additional information without them needing to search through the appendices.

---

### Official Review · Reviewer_gmYS · 2024-11-06

**Soundness:** 2
**Presentation:** 2
**Contribution:** 2
**Rating:** 3
**Confidence:** 4

**Summary:**

The paper introduces a generative benchmark designed to evaluate large language models (LLMs) in the context of hardware design automation. Recognizing the limitations of existing benchmarks focused primarily on basic code generation, GenBen extends evaluation to include critical aspects like Quality-of-Result (QoR) metrics, debugging capabilities, design diversity, and prevention of test set contamination. The benchmark encompasses tasks across multiple design levels, from high-level architecture to low-level circuit optimization, and includes a tiered difficulty system to provide insights into LLM performance across different complexities. With an open-source, end-to-end testing infrastructure, GenBen aims to deliver reproducible and comprehensive assessments, intending to advance the development of LLM-aided design tools within the hardware design community.

**Strengths:**

The paper tries to address a relevant problem and is well written, and well organized

**Weaknesses:**

1)	The dynamic perturbation strategy aims to prevent memorization but may inadvertently introduce ambiguity into test cases. This can create inconsistencies in evaluating LLM performance, especially if slight changes in prompt phrasing lead to variations in model responses. The reliance on superficial perturbations (surface-level changes) may not effectively challenge models in understanding complex circuit design concepts.
2)	The paper identifies issues with synthesizability due to non-IEEE-compliant code in pre-training datasets, but it lacks concrete methods for systematically identifying and filtering these cases. This limitation could lead to significant variability in QoR metrics, particularly for synthesis tools that adhere strictly to IEEE standards, reducing the benchmark’s reliability.
3)	Timing issues are briefly discussed, but the GenBen benchmark does not appear to account for complex timing closure tasks, such as handling Total Negative Slack (TNS) and Worst Negative Slack (WNS) for high-frequency designs. The benchmark could fall short in evaluating LLMs’ ability to generate designs that meet stringent timing requirements, which is critical for high-performance applications.
4)	Although the benchmark includes debugging tests, it may not fully capture the complexity of real-world hardware debugging, particularly for stateful designs or asynchronous circuits. The current debugging scope seems limited to relatively straightforward syntax and functional errors, without addressing state-based errors, race conditions, or setup/hold timing violations that are common in complex designs.
5)	While the inclusion of multimodal tests is innovative, the integration of textual and visual data (e.g., diagrams) lacks specific detail on how visual data is processed or scored. This lack of clarity may lead to ambiguous scoring for models that perform differently across multimodal and text-based tasks, making it challenging to standardize assessments.
6)	The QoR metrics focus on synthesizability, power, area, and timing but do not assess finer aspects like pipeline balancing, parallelism optimizations, or state-machine efficiency. These factors are crucial for high-performance designs and should be part of a rigorous hardware benchmark that targets comprehensive design quality.

**Questions:**

1)	How does GenBen standardize the evaluation of multimodal tests, particularly in handling visual data like circuit diagrams, to ensure consistent scoring and interpretation across different modalities?
2)	Does GenBen incorporate any form of silicon validation, such as FPGA or ASIC implementation, to assess the physical feasibility of LLM-generated designs and identify issues that arise only in physical implementation?
3)	To what extent might GenBen’s reliance on open-source EDA tools impact the accuracy of its QoR metrics and its applicability in industrial-level hardware design tasks where advanced timing and power optimizations are essential?

---

### Official Review · Reviewer_LK5T · 2024-11-08

**Soundness:** 2
**Presentation:** 1
**Contribution:** 2
**Rating:** 3
**Confidence:** 3

**Summary:**

GenBen introduces a benchmark designed to evaluate the capacity of LLMs for generating hardware designs. Existing state-of-the-art benchmarks have several limitations: primarily, they focus solely on syntax and functional pass rates and often include simplistic problems sourced from textbooks, which may be part of LLM training data.

**Strengths:**

In this paper, we present GenBen, a comprehensive benchmark developed to assess the capabilities of LLMs in hardware design. GenBen addresses existing limitations through the following strategies:

1. Incorporating problems sourced from various origins, including silicon-proven projects, to establish three levels of difficulty for evaluating LLMs.
2. Utilizing perturbation strategies to create dynamic tests, ensuring that LLMs encounter unseen challenges.
3. Expanding evaluation criteria beyond syntax and functional pass rates to include metrics such as synthesizability, power, performance, and area.
4. Conducting assessments across multiple LLMs to provide comparative insights.

**Weaknesses:**

The writing quality of this paper is subpar, primarily due to the lack of practical examples. The authors should include more concrete examples to clarify concepts such as "adding some perturbations." Additionally, Table 4 lacks substantive information, and it would be more effective if it included illustrative examples. Furthermore, Section 3.3.1 provides only a superficial overview of the end-to-end verification flow, lacking sufficient detail and making the article less informative.

Moreover, it is unclear how the perturbation process ensures the creation of sound problems, avoiding ambiguity or unclear meanings. The authors should address this aspect to reinforce the reliability of their approach.

Some types on writing:
Section 3.2 "GenBen then generates test tests from the test dataset D using scripts", test typo.
Figure 2 and Figure 4-12 are very hard to read. Use contrasting colors then patterns to discern between different
metrics.
line 213: test tests from

**Questions:**

Please see the weakness section

---

### Meta-Review · Area_Chair_oxAf · 2024-12-21

**Metareview:**

**Summary:**
The paper introduces GenBen, a benchmark designed to evaluate LLMs for hardware design tasks, addressing limitations in existing benchmarks. It includes multi-level tasks, incorporates Quality-of-Result (QoR) metrics, and employs perturbation strategies to prevent test set contamination. GenBen spans diverse tasks, supports multimodal inputs, offers a tiered difficulty system, and demonstrates its utility through evaluations of nine LLMs.

**Strength:**

1. The paper introduces a diverse benchmark that evaluates both functional and QoR metrics, filling in a gap in hardware design benchmarks.

2. The use of perturbation strategies to prevent data contamination is novel and can potentially enhance the benchmark's reliability.

3. The taxonomy of the LAD benchmark provided by this work is potentially insightful and valuable for the research community.

**Weakness:**

1. The perturbation strategy lacks detailed descriptions to avoid ambiguity and lacks thorough validation, raising concerns about its consistency and usability.

2. Some critical aspects of the benchmark design are insufficiently detailed, such as how complex timing metrics are accounted for and how debugging tests capture the complexity of real-world hardware debugging.

3. The paper suffers from some writing issues, including typos and insufficient clarity in figures and methodology, which negatively affect its presentation.


**Reasons for the decision:**

This work aims to provide a comprehensive benchmark encompassing many diverse aspects; however, the methodology and evaluation for each aspect are not sufficiently detailed or validated. Therefore, I am inclined to recommend rejection.

**Additional Comments On Reviewer Discussion:**

The authors did not provide a rebuttal, and most reviewers leaned toward rejecting this paper. I concur with this decision.

---

### Decision · Program_Chairs · 2025-01-22

Reject